# Analyses of emerging macrocyclic lactone resistance: Speed and signature of ivermectin and moxidectin selection and evidence of a shared genetic locus

Jennifer McIntyre[1]*, Alison Morrison[2], Kirsty Maitland[1], Eileen Devaney[1], James A. Cotton[1], Collette Britton[1], Ray M. Kaplan[3], Dave Bartley[2], Roz Laing[1]*

1 School of Biodiversity, One Health and Veterinary Medicine, University of Glasgow, Glasgow, Scotland, United Kingdom, 2 Moredun Research Institute, Pentlands Science Park, Penicuik, United Kingdom, 3 School of Veterinary Medicine, St. George's University, True Blue, Grenada, West Indies

* Jennifer.McIntyre@glasgow.ac.uk (JM); Rosalind.Laing@glasgow.ac.uk (RL)

## Abstract

Subtherapeutic treatment or 'underdosing' is considered a common problem in the control of parasitic helminths of animals and people and can hasten the emergence of anthelmintic resistance. Increasing reliance on the long-acting macrocyclic lactone, moxidectin, in both veterinary and medical settings may increase exposure of incoming helminth populations to subtherapeutic drug concentrations due to its extended half-life. However, we lack genetic markers to monitor emerging resistance as the mechanism(s) underlying resistance to the macrocyclic lactones are unresolved in parasitic helminths. Furthermore, the impact of prior ivermectin exposure on the evolution of moxidectin resistance is unclear. To test the impact of subtherapeutic selection on the emergence of macrocyclic lactone resistance, we exposed a fully drug susceptible isolate of an economically important parasitic helminth of livestock, *Haemonchus contortus*, to low but increasing doses of ivermectin or moxidectin *in vivo* for phenotypic, genomic, and transcriptomic analyses. After a single subtherapeutic dose of ivermectin or moxidectin, we find evidence of selection at a shared genetic locus on Chromosome V, with the signal of selection increasing with subsequent doses. After only three subtherapeutic treatments, ivermectin-selected lines were resistant to a full standard (label) dose of ivermectin. However, moxidectin selected lines remained susceptible to a half dose of moxidectin. This was despite showing higher resistance to ivermectin *in vitro* and a stronger signal of selection at the Chromosome V locus than the equivalent ivermectin-selected lines. Our findings highlight the rapid selection for anthelmintic resistance with subtherapeutic treatment and implicate the pre-existence of ivermectin and moxidectin resistance haplotypes in a drug-naïve population. We demonstrate that ivermectin selected lines show emerging moxidectin resistance, underpinned by a shared genetic locus of resistance. Finally, we

which permits unrestricted use, distribution, and reproduction in any medium, provided the original author and source are credited.

**Data availability statement:** The data for this study have been deposited in the European Nucleotide Archive (ENA) at EMBL-EBI under accession number PRJEB88928 (https://www.ebi.ac.uk/ena/browser/view/PRJEB88928). Code used in this project is available at https://github.com/SheepwormJM/Subtherapeutic-selection-of-Haemonchus-contortus-with-ivermectin-and-moxidectin.

**Funding:** JM, AM, KM, DB and RL were supported by a Wellcome Clinical Research Career Development Fellowship award to RL [216614/Z/19/Z]. The funder had no role in study design, data collection and analysis, decision to publish, or preparation of the manuscript. https://wellcome.org.

**Competing interests:** The authors have declared that no competing interests exist.

speculate that key differences in the resistance phenotype between ivermectin and moxidectin selected lines relate to differences in the inheritance of resistance within this shared locus, with ivermectin resistance manifesting as dominant trait while moxidectin resistance appears to be recessive.

## Author summary

Parasitic helminth infections in people and animals are treated and controlled with anthelmintic drugs, such as the macrocyclic lactones. Unfortunately, anthelmintic resistance is now widespread in parasitic helminths, particularly those of livestock. Ivermectin and moxidectin are distinct but related macrocyclic lactones and the mechanism(s) of resistance to these anthelmintics is unknown. We show that exposing fully drug susceptible populations of parasitic helminths to low doses of either ivermectin or moxidectin rapidly selects for resistance. We find a shared genetic locus underlying resistance to both anthelmintics but that the level of resistance differs between ivermectin and moxidectin selected populations, suggesting important differences in the genetic basis of resistance to each. Moxidectin resistant populations are known to show high levels of ivermectin resistance, but we also demonstrate that ivermectin resistant populations show emerging resistance to moxidectin. These findings are important for the increasing use of moxidectin in people and animals previously treated with ivermectin, and highlight the risk of subtherapeutic dosing, which is thought to be common in mass drug administration programmes for parasite control in both medical and veterinary fields.

## Introduction

The macrocyclic lactones are among the most widely used anthelmintics in animal health and human medicine. Ivermectin, an avermectin, and moxidectin, a milbemycin, are closely related but distinct macrocyclic lactones with similar indications [1,2]. Both are highly effective endectocides, used to treat gastrointestinal nematodes, heartworm, and a variety of ectoparasites in animals. Resistance to both compounds is widespread in the veterinary field, although moxidectin is – at least initially – effective in ivermectin resistant populations [3]. However, moxidectin resistant populations are highly ivermectin resistant [4], implicating a shared mechanism of resistance. In humans, ivermectin has been the mainstay of control programmes for onchocerciasis for many decades and moxidectin was recently licensed in the hope of hastening progress towards eradication. However, the impact of previous treatment with ivermectin, particularly in 'sub-optimal responder' *Onchocerca volvulus* populations [5,6], on the efficacy of moxidectin treatment is uncertain.

Moxidectin is an anthelmintic with a long half-life, which provides a period of persistency against re-infection for weeks to months. This reduces treatment frequency,

which is convenient in both veterinary and human medical settings. However, incoming infective larvae will inevitably be exposed to subtherapeutic doses of the anthelmintic for an extended period post-treatment. Further, in sheep farming, moxidectin is a popular treatment for ewes at lambing, with subtherapeutic concentrations of the drug excreted in milk and ingested by lambs [7]. Anthelmintic treatment at subtherapeutic doses ('underdosing') has long been considered a risk factor for the development of anthelmintic resistance [8]. This assumes that in a drug naïve population, the mutation(s) conferring resistance will be present at a very low frequency, so any carriers are likely to be heterozygotes [9]. Exposing the population to a subtherapeutic dose of a drug may allow such heterozygotes to survive treatment. The potential relevance of this model for moxidectin resistance is supported by previous work showing a selective advantage for ivermectin resistant heterozygotes in establishing infection post-moxidectin treatment [9]. It is also possible that in the early stages of resistance emergence, following subtherapeutic exposure to anthelmintics, there could be a role for mechanisms that enhance tolerance, for example upregulation of drug detoxification or drug export proteins; theoretically these could allow helminths to survive low concentrations of drug without the potential fitness costs of mutations affecting gene function [10]. If this was the case, upregulation of drug detoxification or export proteins might be expected to confer cross resistance to unrelated drug classes. The relevance of these two scenarios in the emergence of macrocyclic lactone resistance has not been investigated to date.

In this study, we exposed a fully drug susceptible isolate of *Haemonchus contortus,* a gastrointestinal nematode of sheep, to subtherapeutic but increasing doses of either ivermectin or moxidectin *in vivo* for three generations. The parental population was used to establish nine separate lines: three under ivermectin selection, three under moxidectin selection and three untreated control lines. Pre- and post-treatment faecal samples were collected from donor sheep at each generation and used for faecal egg counts (FECs) and coproculture to generate infective third stage larvae ($L_3$). Genomic DNA was isolated from pools of 200 post-treatment $L_3$ from three generations of every line for genomic analyses. Adult worms from the third generation of selection were used for RNA-sequencing. Larval development assays were undertaken pre- and post-treatment in the third generation of selection to measure resistance to ivermectin, and test for cross-resistance to benzimidazoles and levamisole. Finally, after the three generations of selection, all lines were subjected to full and half standard (label) doses of ivermectin or moxidectin to test for clinical resistance, and the ivermectin-selected lines were tested for side resistance to moxidectin.

## Materials and methods

### Ethics statement

All experimental procedures were examined and approved by the Moredun Research Institute Experiments and Ethics Committee and were conducted under approved UK Home Office licenses following the Animals (Scientific Procedures) Act of 1986. The Home Office licence number is PP6939295 and the experimental code identifiers are E27/19, E02/22 and E05/23.

### Generation of parasite material

Nine donor sheep were treated with 2.5 mg/kg bodyweight monepantel (Zolvix Oral Solution for Sheep, Elanco AH) 14 days prior to infection to ensure they were parasite free. For the first generation of selection, each donor was orally infected with 5000 $L_3$ of the *H. contortus* fully drug susceptible isolate, MHco3(ISE) [11,12]. On day 28 post-infection, three donor sheep were treated with 0.012 mg/kg body weight (1/16th standard dose) ivermectin (Oramec Drench, Boehringer Ingelheim Animal Health UK Ltd), three were treated with 0.002 mg/kg (1/100th standard dose) moxidectin (Cydectin 0.1% Oral Drench, Zoetis), and three were left untreated. The dosage administered was chosen to achieve a target reduction in the faecal egg output of ~95% [13,14]. Faecal egg counts (FECs), with a sensitivity of up to 1 egg per gram, and coprocultures (to generate infective $L_3$) were performed before and after treatment [15,16]. Adult worms were harvested

at post-mortem on day 42 post-infection (14 days post-treatment), sexed and snap frozen in liquid nitrogen for storage at -80°C. For the second and third generations of selection, 5000 $L_3$ from each post-treatment population of the previous generation were used to orally infect nine further donors. The selection experiment proceeded exactly as for the first generation, other than where the treatment doses were increased. In the second generation the doses were 0.012 mg/kg (1/16th standard dose) ivermectin and 0.008 mg/kg (1/25th standard dose) moxidectin. In the third generation these were 0.02 mg/kg (1/10th standard dose) ivermectin and 0.02 mg/kg (1/10th standard dose) moxidectin (S1 Fig).

To test for cross resistance to moxidectin in ivermectin selected lines, three donor sheep were infected with 5000 post-treatment $L_3$ from the third generation of the ivermectin selected lines and three donors were infected with 5000 $L_3$ from the third generation of the control lines. All six donors were treated with 0.002 mg/kg moxidectin (1/100th standard dose). FECs were performed before and after treatment and adult worms were isolated at post-mortem.

To test for full resistance in the selected lines, one donor sheep was infected with 5000 post-treatment $L_3$ pooled from all third generation ivermectin-selected lines and one donor sheep was infected with 5000 post-treatment $L_3$ pooled from all third generation moxidectin-selected lines. Each donor was treated with a full standard dose (0.2 mg/kg) of ivermectin or moxidectin and FECs were performed before and after treatment. To further investigate the differing phenotypes for the ivermectin and moxidectin lines, three donor sheep were infected with 5000 post-treatment $L_3$ from the third generation of the three ivermectin selected lines and treated with a half dose (0.1 mg/kg) ivermectin, while three donors were infected with 5000 $L_3$ from the third generation of the moxidectin lines and treated with a half dose (0.1 mg/kg) moxidectin. FECs were performed before and after treatment and adult worms were isolated at post-mortem.

### Larval development assays

To make the drug plates, stock solutions of ivermectin aglycone (Santa Cruz Animal Health, 202189), thiabendazole (Sigma, T8904), and tetramisole hydrochloride (Sigma, L9756) were prepared and used for doubling dilutions as follows: ivermectin aglycone (stock: 100 µM in DMSO, dilutions: 50 µM to 0.0485 µM), thiabendazole (stock: 1 mM in DMSO, dilutions: 500 µM to 0.06125 µM), tetramisole hydrochloride (stock: 2.5 mM in water, dilutions: 1250 µM to 1.287 µM). Next, 2% agarose was heated in a microwave until completely melted then placed in a 50°C water bath to equilibrate. In a laminar flow cabinet, 198 µl agarose was dispensed into 11 wells of a 96-well flat bottom microplate (Nunc, 167574) in duplicate for each drug concentration and 2 µl of drug dilution immediately added. The agarose and anthelmintic were gently mixed by pipetting and left to set at room temperature, before storing plates at 4°C in a sealed ziplock bag. Drug plates were brought to room temperature on the bench for 1–2 hours before use. Moxidectin was not used in the larval development assay as ivermectin aglycone was found to provide superior discrimination of moxidectin resistance *in vitro* [4].

Fresh faeces were collected ~24 h before starting the assay and kept in anaerobic conditions at room temperature. Eggs were isolated using standard procedures [16] before diluting to 10,000 eggs in 2 mL tap water with 22.5 mg/mL amphotericin B. After vortexing, 20 µl of the suspension (~100 eggs) was added to each well of the drug plate. Plates were sealed with parafilm and incubated at 26°C. After 24 h, 20 µl nutritive media (Earle's salt solution [10% v/v], yeast extract [1% w/v], sodium bicarbonate [1 mM] and saline solution [0.9% sodium chloride w/v]) [17] was added per well. The assay was terminated on day 6 with the addition of Lugol's iodine. The numbers of eggs, $L_1$, $L_2$ and $L_3$ were counted per well using an inverted light microscope.

### Estimating anthelmintic efficacy with FECs and the larval development assay

For the FECs, the efficacy of treatment was estimated using the R package eggCounts [18] with paired data (pre- and post- treatment FECs from each animal) without individual efficacy of the drug, without zero inflation, and without a correction factor. The FEC reduction (FECR) was estimated as the mode of the posterior density distribution, and the 95% confidence intervals were also reported. For the larval development assays, dose response curves and ED50 values (representing the dose that inhibits development in 50% of the population) were calculated in R Studio (version

2022.02.0.443) [19] with drc [20] using the four-parameter log-logistic function (LL.4). Resistance ratios were calculated as the ED50 of the selected line divided by the ED50 of the time matched control line.

## DNA isolation and library preparation for Pool-seq

Pools of 200 $L_3$ from the MHco3(ISE) parental population and from each donor in the three generations of selection at the post-treatment timepoint (28 populations in total) were used for DNA isolation using the Monarch Genomic DNA Purification Kit (New England Biolabs). After isolation, DNA was stored at 4°C before being sent to Liverpool Centre for Genomic Research (CGR) for library preparation and sequencing. Pool-seq libraries were prepared using the NEBNext Ultra II FS DNA Library Prep Kit (New England Biolabs) and sequenced on the Illumina NovaSeq using S4 chemistry to generate 150 bp paired end reads.

## RNA isolation and library preparation for RNA-seq

Adult worms from each donor/line after the third generation of selection (nine populations) were used for RNA-seq. Total RNA was isolated from separate pools of 20 male and 20 female worms per population using a standard Trizol method described previously [21]. Total RNA was stored at -80°C then shipped to Liverpool CGR on dry ice. RNA-seq libraries were prepared using NEBNext PolyA Selection and Ultra Directional RNA Library Prep Kits (Illumina, E7490, E7420) and sequenced on the Illumina NovaSeq using S1 chemistry to generate 150 bp paired end reads.

## Pool-seq alignment to MHco3(ISE) assembly (Hconv4.0)

Trimmed reads were received from Liverpool CGR where Cutadapt v1.2.1 [22] with –O 3 had been used to trim adapter sequences from the 3' end. Sickle v1.200 [23] had been used for quality trimming with minimum window quality score of 20, reads <15 bp removed and singletons discarded. Fastq files were quality checked using FastQC v0.11.8 [24] and MultiQC v1.6 [25]. Reads were aligned to the MHco3(ISE) Hconv4.0 reference assembly [26] hosted on WormBase ParaSite v18 [27,28] using bwa v0.7.17 *mem* (run parameters -Y -M -C) [29,30]. Duplicate reads in the samtools sorted file were identified using Picard v2.5.0-2 *MarkDuplicates* [31] before filtering unpaired and/or unmapped reads using samtools v1.3 *view* [32]. Next, an mpileup file was made removing duplicate reads (run parameters –B –Q30 –q20 --ff DUP), using samtools v1.17. Indels were identified using popoolation2 v1201 [33] *identify-indel-regions.pl*, and five bases surrounding each indel were marked (--indel-window 5 --min-count 2). After syncing the file using popoolation2 *mpileup2sync.jar* (--min-qual 20), the indel gtf file and *filter-sync-by-gtf.pl* were used to filter bases directly surrounding indels, to reduce errors due to read misalignment. For certain analyses where variation in coverage depth could have a bigger impact, a subsampled mpileup was produced. Following assessment of coverage depth (samtools v1.17 *depth* -a -q30 -Q20) across all samples, we chose the minimum depth such that 75% of sites in each sample achieved that depth or greater (57X). Using popoolation2 *subsample-synchronized.pl* (--target-coverage 57 --max-coverage 2% --method withoutreplace), any site which did not have at least 57X coverage in all samples, or was greater than the top 2% of coverage in any sample, was removed from the mpileup, and the mpileup subsampled to an even coverage of 57X, retaining 89.9 M sites (31.7% of bases in assembly).

## Nucleotide diversity and Tajima's D

Using the subsampled 57X synced mpileup file and grenedalf v0.3.0 [34] *diversity,* sample diversity statistics, including nucleotide diversity (θπ) and Tajima's D, were calculated in 100 kb windows along the genome (--filter-sample-min-count 2 --filter-sample-min-coverage 57 --filter-sample-max-coverage 57 --window-type sliding --window-sliding-width 100000 --window-sliding-stride 100000 --pool-sizes 400). Following this, θπ for each window was normalised by dividing by the median θπ for that sample, and the ratio of these normalised values for each sample to that of the parental F0 generation calculated.

### Genetic differentiation (Fst)

Using the subsampled 57X mpileup file, samples were compared using grenedalf v0.3.0 *fst* calculating the genetic differentiation in 10 kb windows along the genome (--filter-sample-min-count 2 --filter-sample-min-coverage 57 --filter-sample-max-coverage 57 --window-type sliding --window-sliding-width 10000 --window-sliding-stride 10000 --method unbiased-nei --pool-sizes 400). Windows that had no SNPs in any pairwise comparison were removed for all $F_{st}$ calculations. Results were plotted and a significance threshold equivalent to five standard deviations above the mean $F_{st}$ across the three F3:F0 comparisons calculated for each anthelmintic.

### Cochran-Mantel-Haenszel (CMH) test

To test whether the same, or different, variants were under selection by ivermectin or moxidectin in each line, a Cochran-Mantel-Haenszel (CMH) test was performed. After producing the complete mpileup file as described above, popoolation2 v1201 *cmh-test.pl (*--min-count 5 --min-coverage 10 --max-coverage 2%*)* was used to test for differences in allele frequency for each individual SNP between timepoints across lines; the 'Code version' script (https://sourceforge.net/p/popoolation2/code/HEAD/tree/; last updated January 2015) was used. This version generates a log10(pvalue) and the log-odds ratio for each SNP. For all tests, three or more pairwise comparisons were included. Briefly, four tests were performed: (1) control F3 samples vs ivermectin selected F3 samples (3 comparisons), (2) control F3 samples vs moxidectin selected F3 samples (3 comparisons), (3) a random control sample (any generation) vs either ivermectin or moxidectin F3 samples (6 comparisons), and (4) selected F2 samples vs selected F3 samples of the same line for ivermectin and moxidectin (6 comparisons). To avoid pseudo-replication, we randomly selected 6 of the 9 available control samples to include in the combined ivermectin and moxidectin comparison. Following the CMH analyses, results were corrected using either the Bonferroni correction (alpha = 0.01, 0.05) or the Benjamini-Hochberg, using a 5% false discovery rate, and compared. Due to the high number of significant SNPs remaining following each correction, an alternative approach was used to identify those most likely to be of interest. SNPs were ranked by p-value, and those in the lowest 1% of p-values were filtered to include only those with a log-odds ratio of less than one (indicating a reduction in the reference allele in F3 selected samples relative to the comparator samples). The proportion of SNPs retained was plotted in 500 kb windows along the genome using *ggplot2* v3.4.2.

### Allele frequency change over time

In drug-selected lines, we expect to select against susceptible alleles, with an increasing frequency of alternative alleles at each subsequent generation following treatment. To investigate this, the reference allele counts were obtained for the region spanning 30–45 Mb on Chromosome V (identified as the major region under drug selection, based on the $F_{st}$ and CMH analyses above), using grenedalf v0.3.0 *frequency* (--write-sample-counts), using as input the complete sync file made with popoolation2 v1201. As grenedalf outputs the counts for the reference allele and the most common alternative allele only, the depth per site for all four possible alleles plus indels was obtained from the mpileup file. To assess the change in the reference allele frequency for each line across the generations from F0 to F3, two linear mixed effects models were fitted using library *lme4* v1.1-36 [35] in R v4.4.2 for each SNP with non-zero coverage in all relevant samples and with reference allele frequency < 0.9 in at least one of the F3 selected samples. Models tested for a significant interaction between treatment and the slope of the regression of reference allele frequency against generation, including replicate line as a random effect (n = 9). Only a single F0 sample was available, therefore, to generate replicates the allele counts for the F0 sample were copied and their influence on the model proportionally reduced. Three comparisons were tested (ivermectin vs control, moxidectin vs control and selected lines vs control lines). The full model (ref_freq ~ treatment*scaled_timenum+(1|replicate)) and nested model omitting the interaction (ref_freq ~ treatment+scaled_timenum+(1|replicate)) were compared using a likelihood ratio test. A significant p-value

indicated rejection, for that SNP, of the null hypothesis that treatment had no effect on the rate of change in reference allele frequency across generations. The slope coefficient of the treatment line was extracted and plotted to estimate the direction of change in reference allele frequency. P-values were adjusted using Holm's correction [36], and SNPs subsequently filtered using an alpha value of 0.001.

A distinct analysis was also performed using the subsampled mpileup file to identify sites with consistent reductions in reference allele count in each generation and in all lines, but that did not similarly reduce from F0 to F3 in any control line. It is possible that strong selection occurred between earlier generations without further selection later and these sites would not fit the pattern of F0 > F1 > F2 > F3. Therefore, in addition to these, sites which had a reference allele count of zero in at least one F3 selected sample for each anthelmintic were identified, with the requirement that the reference allele had to be >=20 in all control F3 samples and the F0 sample. Data was manipulated and plotted using reshape2 v1.4.4 [37], ggplot2 v3.5.1 [38] and patchwork v1.3.0 [39] in R v4.4.2.

## Microhaplotype analysis

Microhaplotypes are short clusters of SNPs which can be determined from aligned reads [40]. Using the nine F3 generation samples, we looked to see whether one or more microhaplotypes were under selection by ivermectin and moxidectin. BAM files were filtered to remove reads with MAPQ <20, or which had an unmapped pair, or were considered to be PCR or optical duplicates (samtools *view* -F 1036, -q20). Grenedalf *frequency* was provided with the complete mpileup (described above) to obtain SNP positions. Two single-copy genes were selected from opposite ends of the Chromosome V locus, and mhFromLowDepSeq [40] *calc_mh_freq.py* (-pool -w 125 -ms 76) script used to calculate allele frequencies for 125 bp loci, allowing up to 76 SNPs per locus (see Github code for rational). Microhaplotype allele frequencies of >0.01 were retained and results plotted in R as previously.

## RNA-seq analysis

Trimmed reads were received from Liverpool CGR and quality checked as described above for the Pool-seq analyses, aligned to the MHco3(ISE) Hconv4.0 reference assembly [26] using STAR [41] and counts for each coding sequence were generated with featureCounts [42]. Differential gene expression was estimated with DESEQ2 v1.34.0 [43] following the standard workflow in the vignette, with alpha of 0.01. RNAseq data was plotted with ggplot2 v3.4.2 and KaryoplotR v1.20.3 [44].

## Results and discussion

1. *Sub-therapeutic treatment (underdosing) rapidly selects for ivermectin and moxidectin resistance with a measurable reduction in drug efficacy after a single dose*

Subtherapeutic treatment (underdosing) was used to select for macrocyclic lactone resistance in a fully susceptible isolate of *H. contortus* (MHco3.ISE), by establishing nine lines (three control lines with no treatment, three lines selected with ivermectin, and three lines selected with moxidectin) from the same parental population. Anthelmintic doses were chosen based on previous literature [13,14] with a target FECR of ~95% in the first generation to apply drug selection while allowing enough adult worms to survive to maintain the lines. Based on faecal egg counts, the first subtherapeutic dose of ivermectin (0.012 mg/kg; 1/16th standard dose) or moxidectin (0.002 mg/kg; 1/100th standard dose) both achieved high efficacy, with FECRs (and 95% confidence intervals) of 98.8% (98.6, 98.9) and 97.4% (97.2, 97.6) respectively (Table 1 and S2 Fig). In contrast, the efficacies of both compounds were significantly reduced by the second generation (i.e., in the offspring of the survivors), with FECRs of 28.1% (23.4, 32.1) for ivermectin (0.012 mg/kg; 1/16th standard dose) and 59.2% (57.2, 60.9) for moxidectin (0.008 mg/kg; 1/25th standard dose). By the third generation, following treatment with 0.02 mg/kg (1/10th standard dose) for both compounds, the FECRs were 38.1% (36.3, 39.8) for ivermectin and 39.8% (38.0, 41.3)

**Table 1. Treatment efficacy and faecal egg count reduction (FECR) from three generations of selection with ivermectin or moxidectin using pre- and post-treatment faecal egg counts from each donor.**

| Generation | Line | Treatment | Treatment Efficacy (%) | Faecal egg count reduction (%) [95% confidence intervals] |
|---|---|---|---|---|
| 1 | IVM1 | 16th (0.012 mg/kg) IVM | 99.8 | 98.8 [98.6, 98.9] |
| 1 | IVM2 | 16th (0.012 mg/kg) IVM | 98.6 | |
| 1 | IVM3 | 16th (0.012 mg/kg) IVM | 98.5 | |
| 1 | MOX1 | 100th (0.002 mg/kg) MOX | 96.5 | 97.4 [97.2, 97.6] |
| 1 | MOX2 | 100th (0.002 mg/kg) MOX | 98.8 | |
| 1 | MOX3 | 100th (0.002 mg/kg) MOX | 96.8 | |
| 2 | IVM1 | 16th (0.012 mg/kg) IVM | 12.7 | 28.1 [23.4, 32.1] |
| 2 | IVM2 | 16th (0.012 mg/kg) IVM | 31.2 | |
| 2 | IVM3 | 16th (0.012 mg/kg) IVM | 43.9 | |
| 2 | MOX1 | 25th (0.008 mg/kg) MOX | -21.1 | 59.2 [57.2, 60.6] |
| 2 | MOX2 | 25th (0.008 mg/kg) MOX | 71.5 | |
| 2 | MOX3 | 25th (0.008 mg/kg) MOX | 63.6 | |
| 3 | IVM1 | 10th (0.02 mg/kg) IVM | 18.1 | 38.1 [36.3, 39.8] |
| 3 | IVM2 | 10th (0.02 mg/kg) IVM | 51.1 | |
| 3 | IVM3 | 10th (0.02 mg/kg) IVM | 34.8 | |
| 3 | MOX1 | 10th (0.02 mg/kg) MOX | 34.1 | 39.8 [38.0, 41.3] |
| 3 | MOX2 | 10th (0.02 mg/kg) MOX | 49.6 | |
| 3 | MOX3 | 10th (0.02 mg/kg) MOX | 31.8 | |

for moxidectin. This rapid decrease in drug efficacy, despite maintaining or increasing the drug dose, was also observed in the adult worms recovered at post-mortem, where few adults survived the first treatment, but hundreds survived subsequent treatments.

As a more sensitive measure of drug efficacy than faecal egg counts, larval development assays were used to calculate the ED50 for development from eggs to $L_3$ on ivermectin plates. Assays were performed using eggs collected from the third generation, pre- and post-treatment. All lines selected with ivermectin or moxidectin showed a shift of the dose response curve to the right compared with controls (Fig 1), with mean ED50 values of 2.68 nM and 3.52 nM (pre- and post-treatment time points respectively) for controls increasing to 37.59 nM (ivermectin selected lines) and 83.80 nM (moxidectin selected lines) post treatment (Table 2). Resistance ratios of 4.5 - 11.2 (ivermectin selected lines) or 8.8 - 19.6 (moxidectin selected lines) pre-treatment, increased to 7.0 - 16.9 (ivermectin) and 11.4 - 42.8 (moxidectin) post treatment. This demonstrates ivermectin resistance in all selected lines and is consistent with the observation that moxidectin resistant populations show side resistance to ivermectin [4,13]. The findings are comparable with published ED50 values and resistance ratios for ML resistant field populations of *H. contortus* in the Southern United States [4]. In the US study, ED50 values were 2.5 nM for a drug susceptible population and 18.4 nM for an ivermectin resistant but moxidectin naïve population, with a resistance ratio of 5.3. For field populations with 'emerging' to 'resistant' moxidectin phenotypes, ED50 values ranged from 19.8 nM to 396.6 nM with resistance ratios from 10.7 to 128.0.

The loss of ivermectin efficacy in all selected lines was in contrast with the results for equivalent assays testing sensitivity to drugs from two different anthelmintic classes, the benzimidazoles and levamisole (S3 Fig and S3 Table), where there was no evidence of cross selection for resistance to either. This is consistent with the expected independent mechanisms of resistance for each anthelmintic class [45], but also suggests that multi-drug resistance mechanisms, which we speculated could play a role in the initial stages of resistance evolution, are not relevant for emerging resistance to the macrocyclic lactones or at least are not observed with this experimental design.

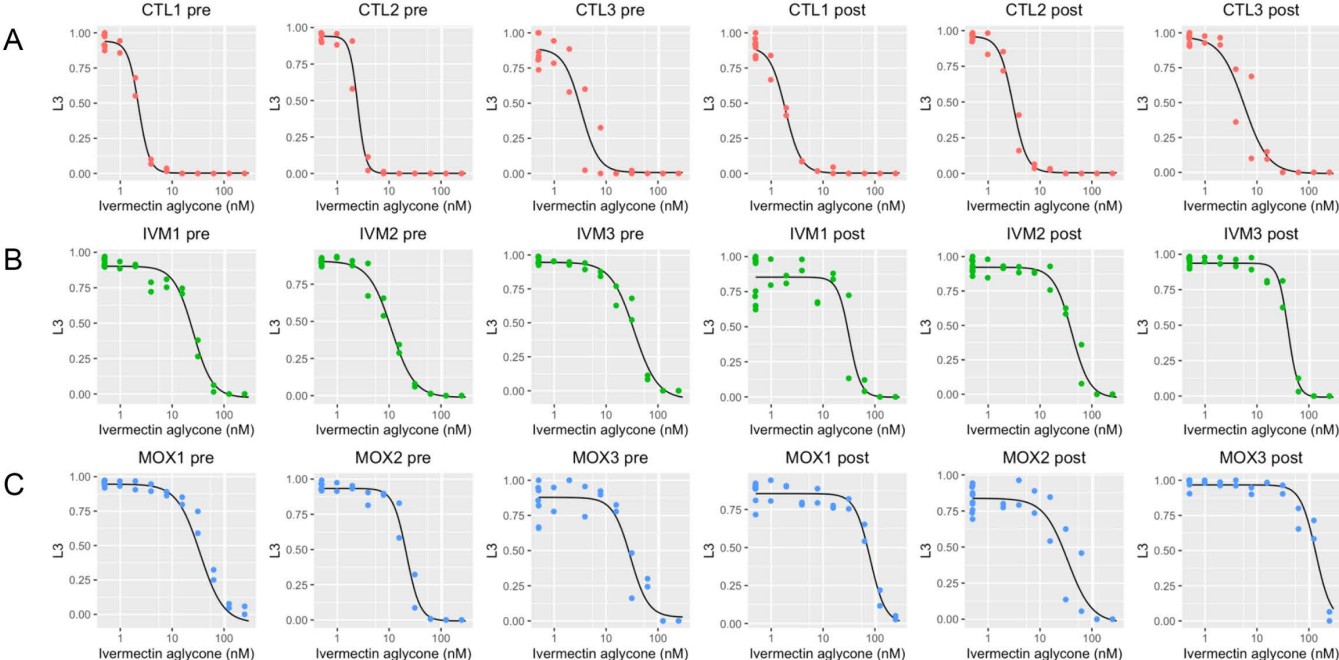

**Fig 1. Larval development on ivermectin aglycone plates.** Larval development assays were undertaken for all lines, at pre- and post-treatment time-points, in the third generation of selection. (A) Control (CTL) lines had no treatment applied but sampling was time-matched with the selected lines ((B) IVM = ivermectin and (C) MOX = moxidectin). Y axis shows the proportion of the population in each well that developed to L3.

**Table 2. ED50s and resistance ratios (RR) from larval development assays on ivermectin plates.**

| Generation | Line | Pre-treatment ED50 | Post-treatment ED50 | Pre-treatment RR | Post-treatment RR |
|---|---|---|---|---|---|
| 3 | CTL1 | 2.26 | 1.89 | – | – |
| 3 | CTL2 | 2.47 | 3.01 | – | – |
| 3 | CTL3 | 3.31 | 5.67 | – | – |
| 3 | IVM1 | 25.36 | 32.00 | 11.22 | 16.93 |
| 3 | IVM2 | 11.09 | 40.93 | 4.49 | 13.60 |
| 3 | IVM3 | 34.69 | 39.83 | 10.48 | 7.02 |
| 3 | MOX1 | 44.33 | 80.84 | 19.62 | 42.77 |
| 3 | MOX2 | 21.71 | 34.26 | 8.79 | 11.38 |
| 3 | MOX3 | 29.58 | 136.29 | 8.94 | 24.04 |

2. *Lines selected with ivermectin showed emerging moxidectin resistance, an important consideration for moxidectin use in ivermectin resistant populations.*

Moxidectin resistant populations are known to be highly resistant to ivermectin [4] and a loss of ivermectin efficacy with subtherapeutic moxidectin treatment was demonstrated with the larval development assays in the current study (Fig 1). However, moxidectin remains initially effective in ivermectin resistant populations [1,14,46] and has been approved for use in sub-optimal responder populations of *Onchocerca volvulus* after long-term ivermectin treatment [6,47]. Therefore, the impact of prior selection with ivermectin on the emergence of moxidectin resistance is an important variable. To explicitly test this, six donor sheep were infected with the three ivermectin selected lines (progeny from the third generation post

treatment) and three time-matched control lines, then treated with a subtherapeutic dose of moxidectin (0.002 mg/kg; 1/100th standard dose) to compare drug efficacies (Table 3 and S4 Fig). Treatment of donors infected with the control lines had a FECR of 90.7% (89.4, 91.7), but treatment of the ivermectin selected lines had zero efficacy; FECR 0% (0, 0). This result confirms that ivermectin treatment primes the population for emerging (low dose) moxidectin resistance, and is an important consideration regarding the increasing use of moxidectin in populations after previous treatment with ivermectin, in both medical and veterinary settings.

3. *After three subtherapeutic treatments, ivermectin-selected lines became resistant to a full dose of ivermectin, but moxidectin-selected lines remained susceptible to a half dose of moxidectin*

To test the relevance of a reduced efficacy of subtherapeutic treatment on clinical resistance, a single donor sheep was infected with $L_3$ pooled from the third generation ivermectin selected lines and treated with a full dose of ivermectin (0.2 mg/kg). A second donor sheep was infected with $L_3$ pooled from the third generation moxidectin selected lines and treated with a full dose of moxidectin (0.2 mg/kg). For the ivermectin-treated donor, the efficacy was 73%, consistent with clinical ivermectin resistance in the pooled population of ivermectin selected lines. In contrast, moxidectin treatment cleared the infection with the moxidectin selected lines. To further investigate this difference in phenotype, six donor sheep were infected with the six third-generation selected lines and treated with a half dose (0.1 mg/kg) of ivermectin or moxidectin. All three ivermectin selected lines were resistant to a half dose of ivermectin with a FECR of 0% (0, 0.01) (Table 4 and S5 Fig) and with hundreds of adult worms harvested at post-mortem in every line. In contrast, a half dose of moxidectin retained high efficacy in all three moxidectin selected lines with a FECR of 99.9% (99.7, 100.0) and with only one adult worm found in one sheep at post-mortem. The difference in the clinical manifestation of resistance between the ivermectin and moxidectin selected lines, despite the moxidectin lines showing equal or higher ivermectin resistance *in vitro* (Fig 1) is striking. Early studies on the inheritance of macrocyclic lactone resistance in *H. contortus* also found notable differences, with moxidectin resistance manifesting as a semi recessive trait [9] in contrast to ivermectin resistance which appeared to be semi dominant [48,49]. However, the genetic variant(s) underlying resistance to either drug have yet to be resolved.

**Table 3. Treatment efficacy and faecal egg count reduction (FECR) after treatment of ivermectin-selected and control lines with moxidectin. Negative values indicate zero efficacy.**

| Generation | Line | Treatment | Treatment Efficacy (%) | Faecal egg reduction (%) [95% confidence intervals] |
|---|---|---|---|---|
| 4 | CTL1 | 100th (0.002 mg/kg) MOX | 82.74 | 90.7 [89.4, 91.7] |
| 4 | CTL2 | 100th (0.002 mg/kg) MOX | 96.98 | |
| 4 | CTL3 | 100th (0.002 mg/kg) MOX | 53.57 | |
| 4 | IVM1 | 100th (0.002 mg/kg) MOX | -210.95 | 0 [0, 0] |
| 4 | IVM2 | 100th (0.002 mg/kg) MOX | 7.30 | |
| 4 | IVM3 | 100th (0.002 mg/kg) MOX | -305.88 | |

**Table 4. Treatment efficacy and faecal egg count reduction (FECR) after treatment with a half dose of ivermectin or moxidectin. Negative values indicate zero efficacy.**

| Generation | Line | Treatment | Treatment Efficacy (%) | Faecal egg reduction (%) [95% confidence intervals] |
|---|---|---|---|---|
| 4 | IVM1 | 1/2 (0.1mg/kg) IVM | 69.57 | 0 [0, 0.01] |
| 4 | IVM2 | 1/2 (0.1mg/kg) IVM | -3.02 | |
| 4 | IVM3 | 1/2 (0.1mg/kg) IVM | -85.71 | |
| 4 | MOX1 | 1/2 (0.1mg/kg) MOX | 99.75 | 99.9 [99.7, 100.0] |
| 4 | MOX2 | 1/2 (0.1mg/kg) MOX | 100.00 | |
| 4 | MOX3 | 1/2 (0.1mg/kg) MOX | 99.93 | |

4. *Genomic signatures of ivermectin and moxidectin selection highlight a shared QTL on Chromosome V implicating a common mechanism of resistance.*

In *H. contortus*, ivermectin resistance is conferred by a quantitative trait locus (QTL) at ~37.5 Mb on Chromosome V based on three genetic crosses between resistant isolates and a susceptible isolate [45,49] and supported by genome-wide studies of global field populations [50–52]. For moxidectin, very little is known about the genetic basis of resistance, and the genetic architecture of emerging resistance is unknown for either compound.

Whole genome sequencing was generated from pools of 200 $L_3$ from the parental population and from each line at each generation post-treatment. As shown in the $F_{st}$ plots of pairwise comparisons between the parental population and post-treatment populations (Fig 2A–2C), a broad region of genetic differentiation over the Chromosome V QTL was apparent in all lines selected with either ivermectin or moxidectin. Strikingly, selection at this locus was apparent after a single treatment with either compound and was particularly marked for the moxidectin lines, with no evidence of selection elsewhere in the genome. Pairwise comparisons of the third generation ivermectin and moxidectin selected lines (Fig 2D) showed little differentiation over the same region, implying that shared haplotypes are under selection by both compounds. This was supported by the Cochran-Mantel-Haenszel test, a test used to assess whether the same SNPs are under selection in multiple independent tests, which also narrowed the QTL to ~36 – 45 Mb (Fig 3). Despite applying selection pressure that was strong enough for only a small subset of adult worms to survive the first treatment, i.e., a severe population bottleneck, nucleotide diversity (θπ) was maintained genome wide in all selected lines relative to the F0 population (S6 Fig). However, on Chromosome V, there was a localised loss of nucleotide diversity between ~36 – 45 Mb, particularly in two moxidectin selected lines; one showed evidence of this by the second generation, the other by the third (Fig 4B). Although the same pattern was seen in the ivermectin selected lines, it was less marked (Fig 4A) and did not occur in the control lines (Fig 4C). Intriguingly, localised increase in nucleotide diversity was apparent either side of this locus (~32.5-36 Mb and ~45-48.5 Mb), and in the centre of the locus at 41 Mb, for all selected lines relative to the F0. This localised increase occurred in regions of particularly low nucleotide diversity in F0 (Fig 4D; 31–34 Mb, 41 Mb and 45–48 Mb) and may relate to selection of one or more initially rare divergent haplotypes early in the experiment.

Tajima's D is a measure estimating the difference between the number of segregating sites and the number of pairwise differences between DNA sequences [53]. Directly following a selection event, dependent on the severity of such an event, there may be very few rare alleles and Tajima's D may be positive. Over time, as new mutations occur, rare alleles will increase and Tajima's D will become negative. As the population interbreeds, the rare alleles will become more common, and Tajima's D values will return towards zero. In the F0 population, the regions of low nucleotide diversity on Chromosome V (Fig 4D) corresponded to negative Tajima's D (Fig 5D). This suggests that the current standing variation within the sample is lower than expected at these loci for the population size. This is likely to be a signature of the bottleneck associated with generating the MHco3(ISE) line (ISE = Inbred Susceptible Edinburgh) [11]. It is notable that all control samples show the same pattern of Tajima's D as the F0 sample (Fig 5C and 5D), unlike the selected lines. In contrast to these regions of negative Tajima's D, the locus from 36-45 Mb shows either neutral or mildly positive Tajima's D in the F0 sample, control lines and F1 generations of the selected lines (Fig 5). However, as selection progresses, and nucleotide diversity reduces, Tajima's D becomes neutral and then negative across this locus in both the ivermectin and moxidectin lines (Fig 5A and 5B).

Overall, these data suggest that sub-therapeutic treatment with either ivermectin or moxidectin has selected pre-existing but rare haplotypes in the QTL on Chromosome V. They also suggest that selection has occurred on the same few haplotypes in all lines, and with both anthelmintics, but that the extent to which this has occurred has varied. For ivermectin, although continued selection was observed with increasing ivermectin concentration, it was less strong than for moxidectin; this is consistent with the resistant phenotype observed after three generations and suggests that sub-therapeutic selection even at very low doses was sufficient to select for full ivermectin resistance in this population. In contrast, for moxidectin, continued selection was observed with increasing moxidectin concentration, and despite showing resistance to 1/10th of a dose, these worms were not resistant to a full or half dose. These findings could suggest a

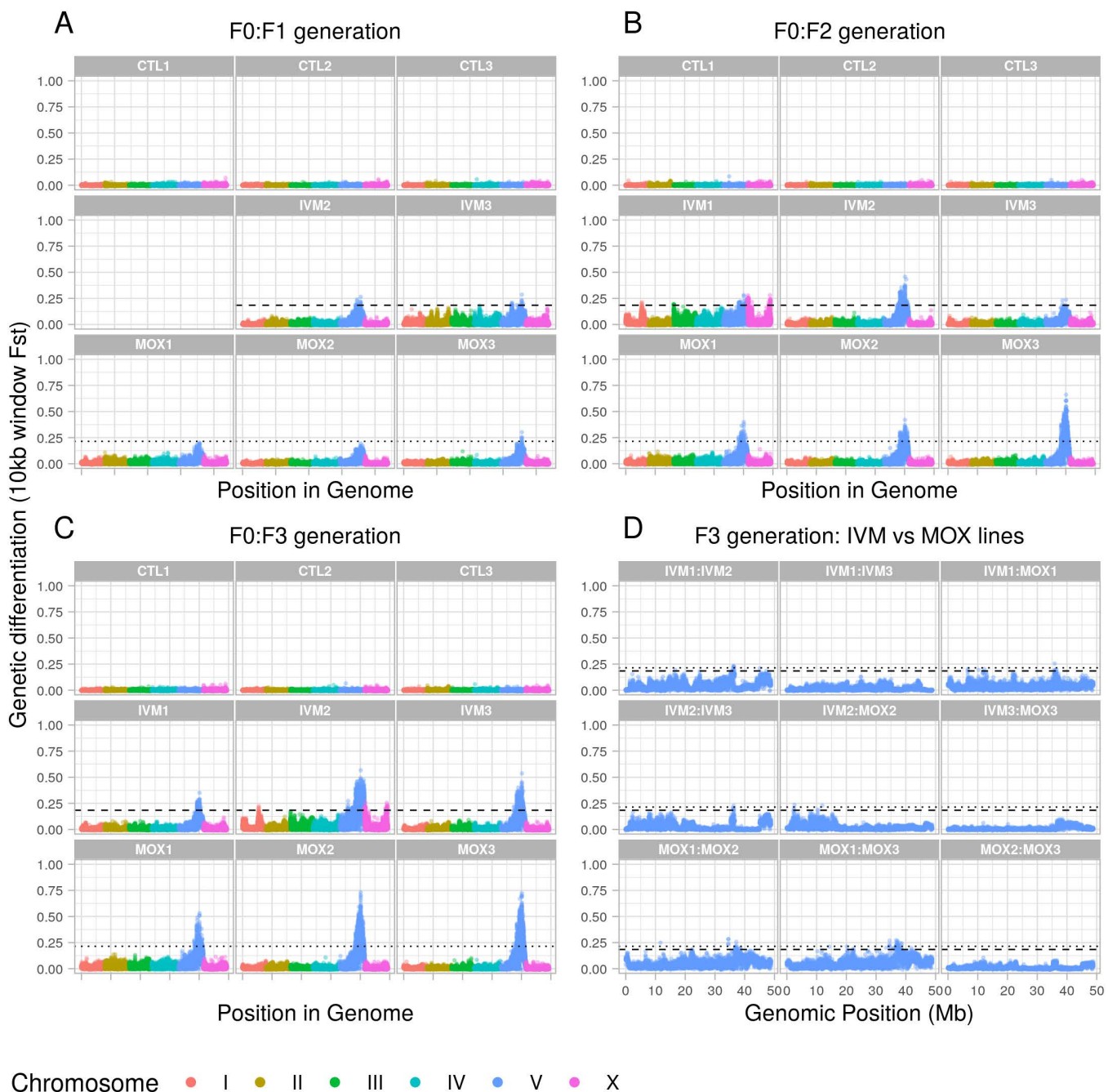

**Fig 2. Pool-seq $F_{st}$ plots from three generations of sub-therapeutic selection.** (A–C) Each sample from each generation is compared with the F0 parental sample and 10 kb window $F_{st}$ values plotted along the chromosomes (I-X). For each of the ivermectin selected (IVM) and moxidectin selected (MOX) lines, a line was plotted indicating five standard deviations above the genome-wide mean of the F3 generations (dashed and dotted respectively). CTL = control lines. Note that no sequence data was available for sample IVM1. (D) Chromosome V pairwise comparisons of third generation ivermectin and moxidectin samples to each other. Horizontal lines are five standard deviations above the mean $F_{st}$ value for the F3 generations – the same as in panels A-C, but with both ivermectin and moxidectin lines overlaid on each faceted plot.

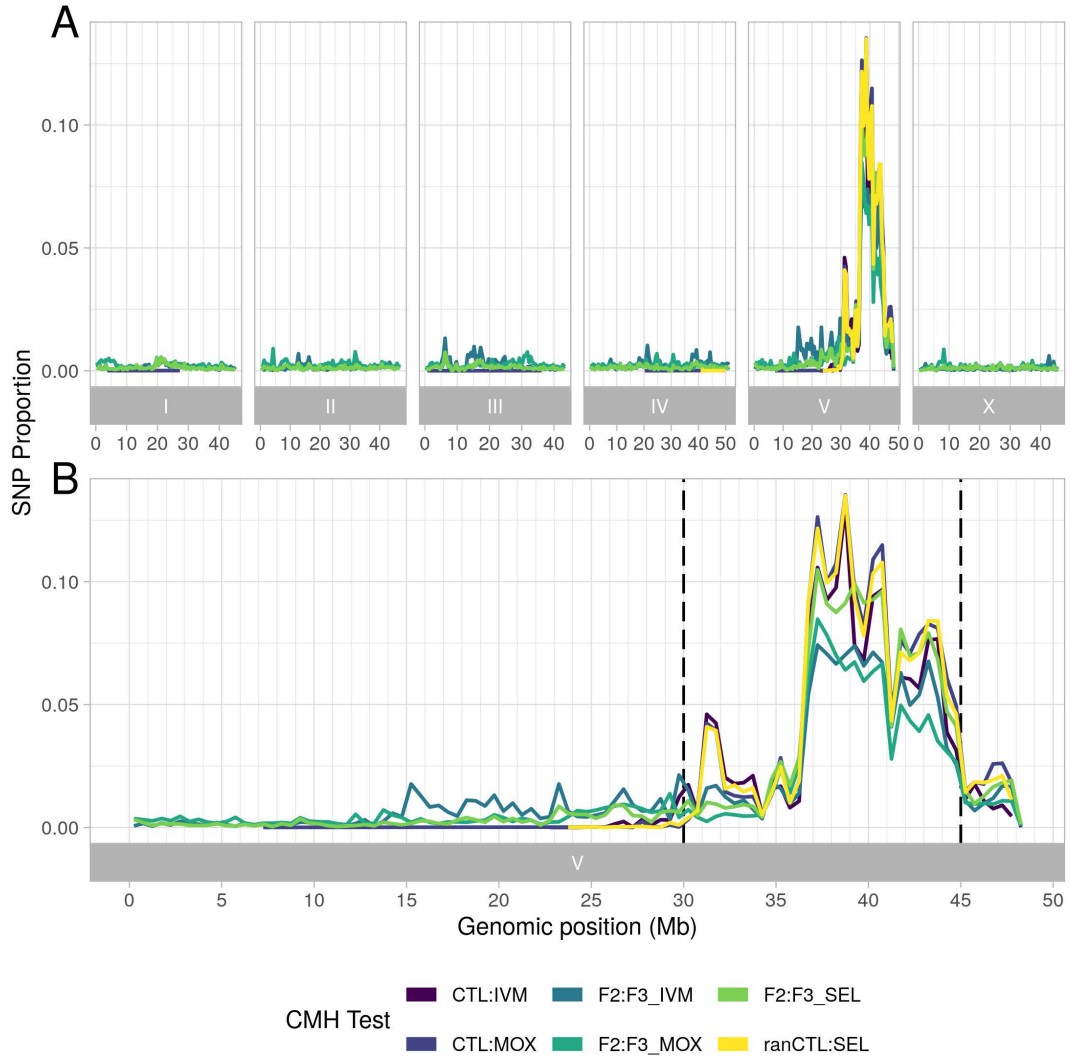

**Fig 3. Cochran-Mantel-Haenszel test showing common selection within and across ivermectin and moxidectin treatment sample groups.** SNP Proportion = Proportion of SNPs per 500 kb window that are in lowest 1% (by rank) of CMH p-values, with a log-odds ratio < 1. All comparisons used the F3 generation of the treated lines. Two also used the F3 generation of the control lines (CTL:IVM and CTL:MOX), three compared the F3 generation with the same F2 treated lines (F2:F3_IVM, F2:F3_MOX and F2:F3_SEL), while a final comparison compared the F3 samples for both treatment groups with a random control sample from any generation (ranCTL:SEL). (A) Whole genome, (B) Chromosome V. Dashed black lines indicate region under selection as found from $F_{st}$ analysis.

multigenic model of resistance, where a single variant (or multiple linked variants) confers resistance to a full dose of ivermectin and partial resistance to moxidectin, but where multiple discrete variants within the Chromosome V locus must be present in an individual worm for resistance to a full dose of moxidectin. They are also consistent with ivermectin resistance manifesting as a dominant trait while moxidectin resistance is recessive.

5. *SNP frequency analyses find initial selection over a broad genomic locus, which is narrowed over subsequent generations and increasing anthelmintic doses*

Despite the use of an inbred parental isolate, and seemingly severe bottlenecking of the population at the first round of selection, over 100 million SNPs were retained across the genome in selected lines, averaging 1 SNP every 2.8 bp. Even

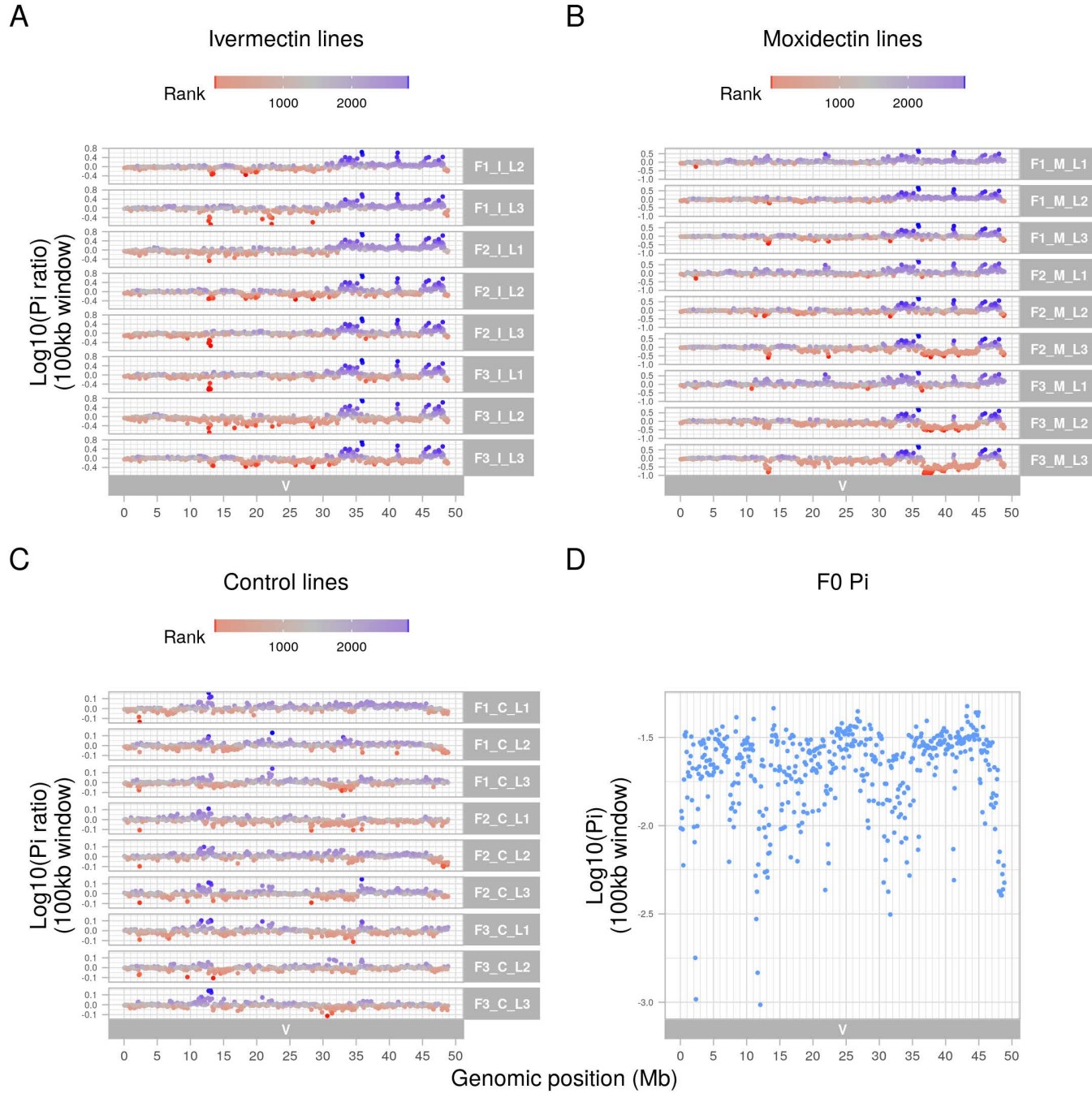

**Fig 4. Nucleotide diversity (θπ) along Chromosome V in all samples calculated in 100 kb windows using a subsampled depth of 57X.** (A–C) Ratio of nucleotide diversity in three generations (F1 to F3) of sub-therapeutic selection in comparison to F0 sample population. Key: C = control, I = ivermectin, M = moxidectin. L1 to L3 indicate biological replicate selection lines. Pi ratio = Normalised Sample θπ/ Normalised F0 θπ. Nucleotide diversity for each sample window was normalised by dividing by the genome-wide median diversity. Colour indicates the rank within the treatment group, where blue indicates a higher diversity than F0, and red a lower diversity. Note that the rank indicates the genome-wide ranking. Note that no sequence data was available for sample F1_I_L1. (D) Nucleotide diversity (θπ) of the F0 sample along Chromosome V. More negative values indicate lower diversity.

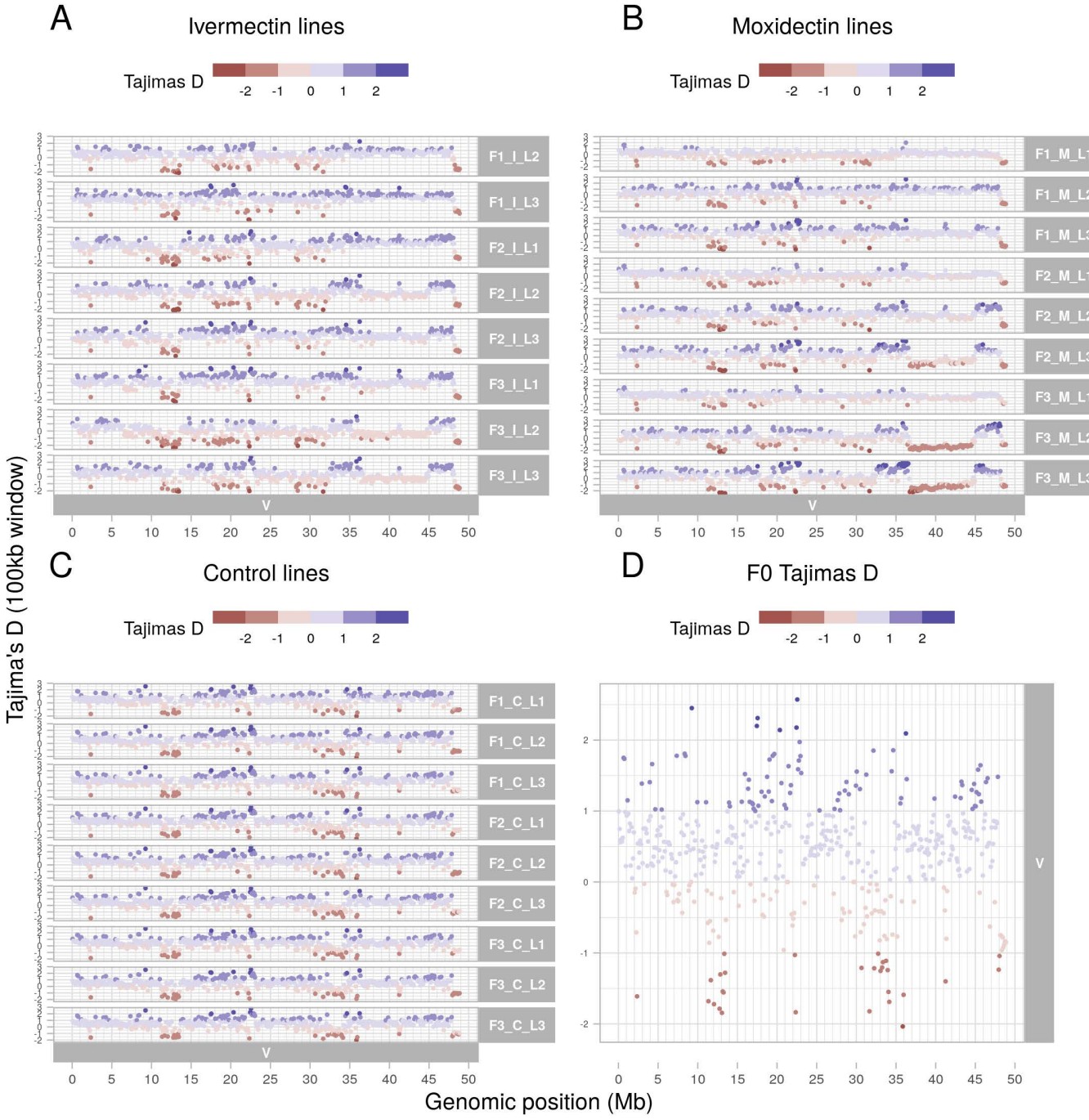

**Fig 5. Tajima's D along Chromosome V in all samples calculated in 100 kb windows using a subsampled depth of 57X.** (A) Ivermectin selected lines, (B) moxidectin selected lines, (C) control lines, (D) F0 sample. Note that no sequence data was available for sample F1_I_L1. More negative values (red) correlate to more rare alleles than expected for the population size, while more positive values (blue) indicate fewer rare alleles than expected and either very few alleles at all (highly conserved), or balancing selection of alleles. If the population is neutrally evolving, Tajima's D is close to zero (mix of rare and common alleles).

with further filtering based on depth and a minimum allele count, between 15.5 to 18 million SNPs were retained genome-wide depending on the samples used for the Cochran-Mantel-Haenzel test (Table 5). Without any depth filtering, 4.6 million SNPs were identified within the 30–45 Mb region of Chromosome V, of which 938,315 SNPs had an alternate allele frequency of at least 0.1.

Considering each SNP independently further clarified selection over the Chromosome V QTL (Fig 6). Using a linear mixed effects-based model, SNPs were identified with a change in reference allele frequency over each subsequent generation from F0 to F3 generations in treated lines relative to control lines. The majority of SNPs within the 30–45 Mb region on Chromosome V did not show marked changes (Fig 6), and many showed an increase in the reference allele over time, some of which were statistically significant. This could mean that one or more of these reference alleles are resistant, or more likely that these represent alternative sensitive alleles segregating within the *Haemonchus* population. However, both ivermectin and moxidectin selection experiments demonstrated significant selection ($p < 0.001$ following Holm's correction) against the reference allele for a considerable subset of SNPs between ~36.5-45 Mb, but particularly over the region from ~36.5-41 Mb (Fig 6A and 6B). Also of note was that the strength of selection over this region (indicated by the slope coefficient) was greater than for the adjacent region from 30-36.5 Mb. Combining both ivermectin and moxidectin lines into a single group, 'selected' and comparing with control lines, identified the same regions, indicating that broadly, shared SNPs are under selection across this QTL by both anthelmintics. A broad region of moderate selection occurred, extending to 20 Mb on Chromosome V in certain lines/samples (Fig 2), yet the Cochran-Mantel-Haenszel results found that continued selection outside of the ~36.5-45 Mb locus did not occur following treatment of the F2 generation for either anthelmintic (Fig 3). Importantly, that selection continued to occur solely within the ~36.5-41 Mb locus from F2 to F3 generations implies that this region contains the variant(s) driving resistance (as expected from previous studies [45,49]) and that recombination is breaking down haplotypes in the wider region. Differences in the pattern of selection between the anthelmintics were apparent. Moxidectin samples demonstrated continued selection over a broader region than ivermectin samples, and a greater reduction in the reference allele for a subset of SNPs between ~36.5-45 Mb for moxidectin lines 2 and 3 than for line 1, or any ivermectin lines (Figs 6A, 6B, and S7). While differences in the pattern of selection between anthelmintics may reflect differences in the dose and potency of each compound, these findings are consistent with moxidectin resistance manifesting as a (more) multigenic and/or recessive trait than ivermectin resistance, and fits with the distinct resistance phenotypes for each drug. In total, 12 SNPs demonstrated a statistically significant, strongly negative slope coefficient of reduction (<-0.5) in the reference allele from F0 to F3 in selected lines. We have included information

**Table 5. Number of SNPs identified as significant by the Cochran-Mantel-Haenzel test.**

| CMH results | FDR 5% (BH) | Bonferroni correction (a = 0.01) | Bonferroni correction (a = 0.05) | SNPs in lowest 1% p-value rank, and which have a log-odds ratio <1; i.e., a reduction in reference allele in selected lines | Total SNPs |
|---|---|---|---|---|---|
| CTL F3 vs IVM F3 (1/10th IVM dose) | 2,884,057 (18.6%) | 493,898 (3.2%) | 539,252 (3.5%) | 40,369 | 15,485,213 |
| CTL F3 vs MOX F3 (1/10th MOX dose) | 3,489,538 (20.0%) | 511,300 (2.9%) | 553,896 (3.2%) | 55,052 | 17,432,827 |
| random CTL F1-F3 vs IVM F3 **&** random CTL F1-F3 vs MOX F3 | 4,819,032 (27.0%) | 817,017 (4.6%) | 885,765 (5.0%) | 53,611 | 17,864,992 |
| IVM F2 vs IVM F3 **&** MOX F2 vs MOX F3 | 227,087 (1.3%) | 21,735 (0.1%) | 27,674 (0.2%) | 73,235 | 18,022,431 |
| IVM F2 vs IVM F3 | ND | ND | ND | 63,836 | 15,602,150 |
| MOX F2 vs MOX F3 | ND | ND | ND | 72,234 | 17,530,120 |

For each test run, the total number of polymorphic sites across all samples used is indicated (Total SNPs), and those considered significant after either FDR or Bonferroni correction presented, with the percentage of total SNPs indicated. The number of those with a reduction in the reference allele in the selected F3 generations relative to the control or F2 samples, which are also in the lowest 1% of SNPs by p-value rank are indicated.

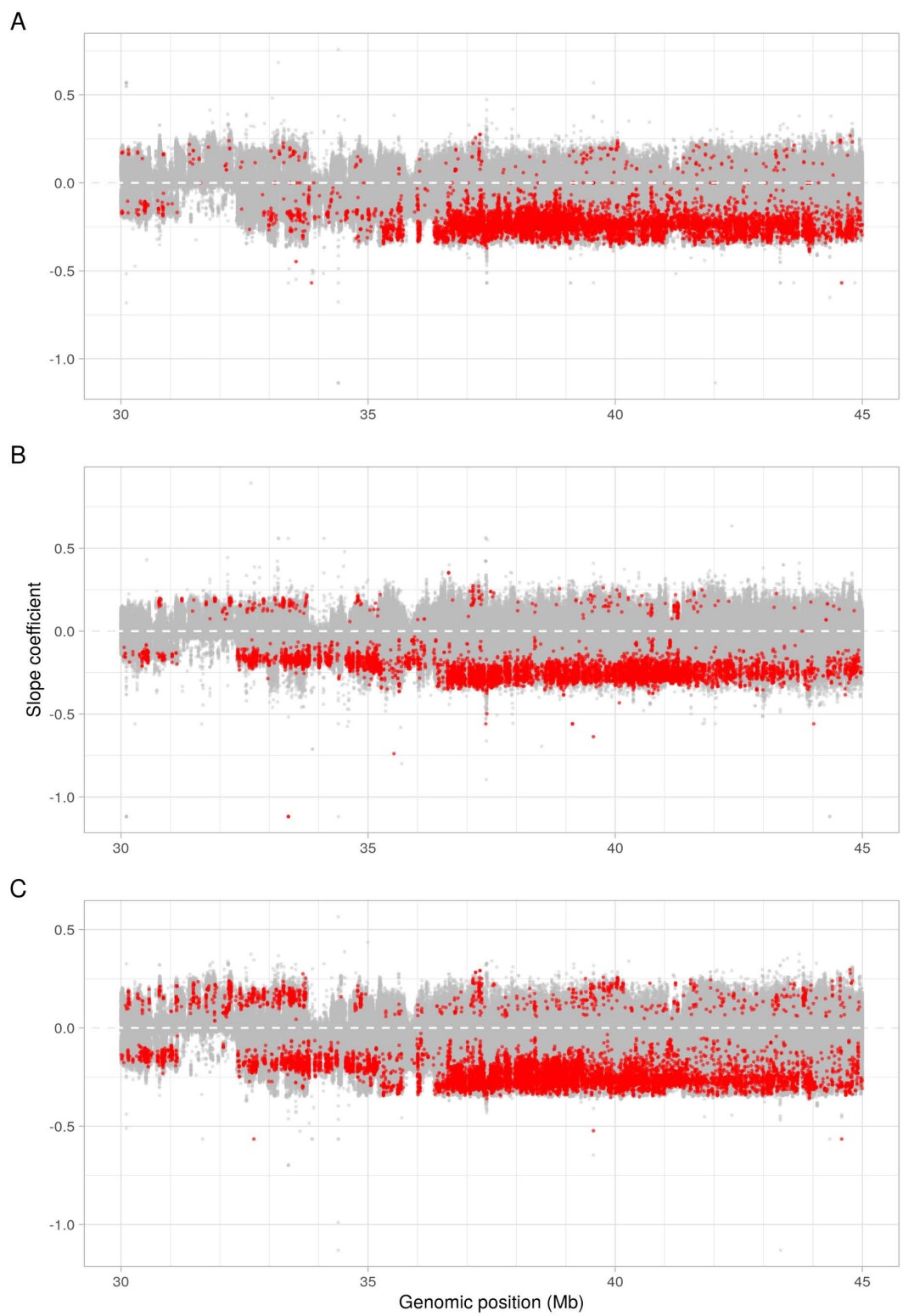

**Fig 6. Change from F0 to F3 in the reference allele frequency of individual SNPs located between Chromosome V:30-45 Mb.** Negative values indicate a reduction in the reference allele frequency at that site from F0 to F3. Each point is a SNP; red points are those which are significantly different to controls following Holm's correction of the p-values, and where the corrected p-value is < 0.001. In other words: if red, the reference allele is changing

in treated lines differently to control lines, and if negative it is reducing in the treated lines. (A) ivermectin vs control lines (slope coefficient plotted for ivermectin lines), (B) moxidectin vs control lines (slope coefficient plotted for moxidectin lines), (C) selected (ivermectin and moxidectin) vs control lines (slope coefficient plotted for combined selected lines).

on these SNPs in S4 Table. Finally, we looked at two single-copy genes located towards either end of the locus, calculating microhaplotype frequencies for these. We found that for each gene, a strong signal of selection was apparent for haplotypes that were distinct from those in control samples (S8 Fig). These haplotypes tended to be the same for both ivermectin and moxidectin selected lines, however the ivermectin selected lines tended to also retain moderate allele frequencies for the main control haplotypes, suggesting a dominant resistance mechanism under selection within the locus. In contrast, moxidectin lines 2 and 3 tended to show almost complete selection against the control haplotypes, suggesting a more recessive resistance mechanism. However, moxidectin line 1, which showed a higher diversity across the locus, often had distinct haplotypes appearing under selection. In addition, it often still showed similar allele frequencies for control haplotypes as the ivermectin lines.

6. *Transcriptomic analysis finds significant dysregulation of gene expression at the Chromosome V locus.*

Previous work identified differential expression of a transcription factor, HCON_00155390 (*cky-1*), encoded at the centre of the Chromosome V locus, in ivermectin resistant isolates [21,45]. In the current study, RNA-seq analysis of male and female adult worms after the third generation of selection found significant dysregulation of gene expression within the same QTL in ivermectin and moxidectin selected lines, relative to controls (Fig 7). In all four pairwise comparisons (treated

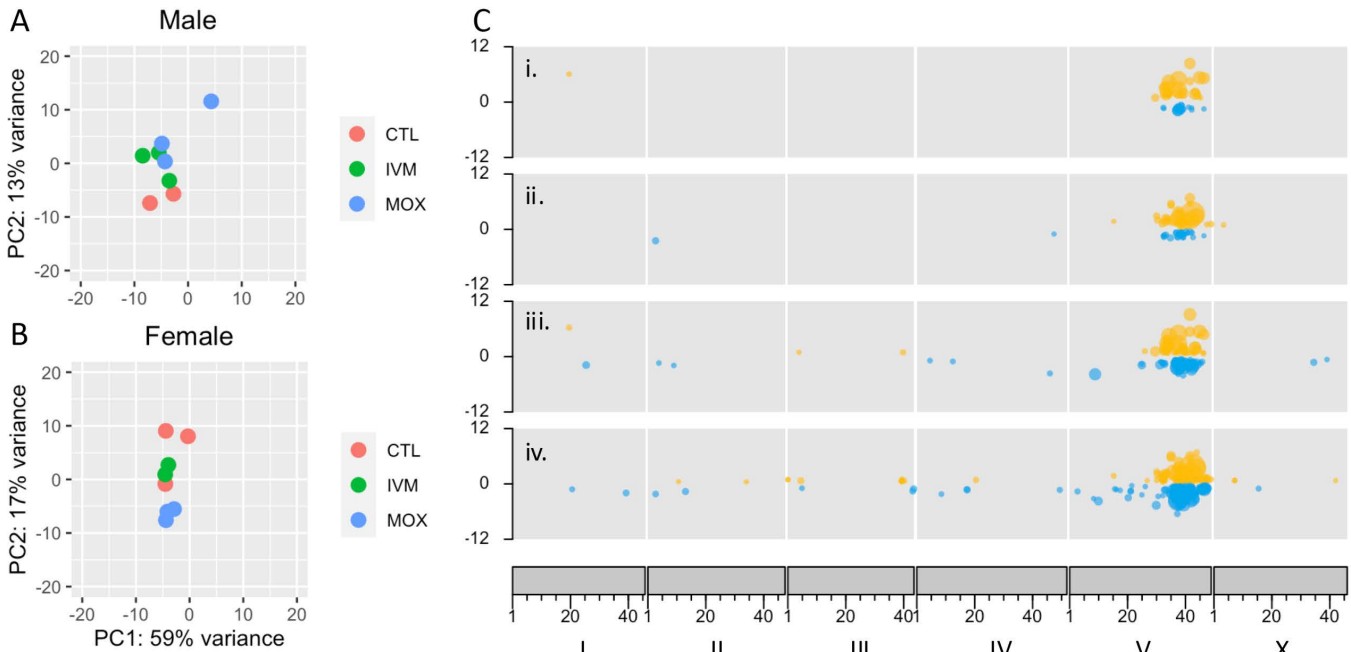

**Fig 7. RNA-seq data for male and female adults after third generation of ivermectin or moxidectin selection.** PCA plots for male (A) and female (B) datasets. Genome-wide differential gene expression visualised as karyoplots (C) with i. ivermectin selected males versus control males, ii. ivermectin selected females versus control females, iii. moxidectin selected males versus control males, iv. moxidectin selected females versus control females. Upregulated genes are in yellow, downregulated genes are in blue, with adjusted P<0.01 for significance.

versus control for males and females for each drug) there were 11 differentially expressed genes in common, all of which were encoded within the Chromosome V locus (Tables 6 and S5). A larger number of differentially expressed genes were identified when the ivermectin treated and moxidectin treated samples were considered independently (42 and 48 differentially expressed genes respectively, relative to controls) but again differential expression was highly localised to genes within the Chromosome V locus. Furthermore, when the ivermectin and moxidectin samples were directly compared to each other, rather than to untreated controls, only three (males) and 14 (females) genes were differentially expressed, and the majority of these were not encoded on Chromosome V. These findings highlight a shared transcriptomic signature of treatment with either ivermectin or moxidectin. Expression of *cky-1* was higher in ivermectin and moxidectin selected males and females than in controls (S9 Fig), but the difference was not statistically significant. There was also no evidence of differential gene expression of multi-drug resistance genes, such as P-glycoproteins, which have been implicated in ivermectin resistance in multiple species of nematode [54–57]. This finding is consistent with the demonstrated lack of cross resistance to different drug classes with this experimental design. Taken together, the transcriptomic data are consistent with one or both of the following: an analogous and highly localised transcriptomic response to treatment with either ivermectin or moxidectin, and/or selection by both drugs of shared resistance haplotypes, with cis-regulated transcriptional profiles that are distinct from those of susceptible haplotypes.

7. *Speed of selection and maintenance of genetic diversity implies pre-existence of multiple resistance variants in a susceptible population.*

Resistance to ivermectin can develop very rapidly in *H. contortus*; early reports of resistance in the field involved two populations of *H. contortus* with a history of only three ivermectin treatments [58]. In the current study, the demonstration of ivermectin and moxidectin selection at the Chromosome V locus, after a single drug exposure in replicated lines of a fully susceptible population, implies the pre-existence of resistance-conferring variants. This is likely to reflect high levels of standing genetic variation in the starting population, despite a history of inbreeding then maintenance in the laboratory since 2004 [11]. Furthermore, the maintenance of genetic diversity surrounding the QTL, after strong selection based on low adult worm survival, i.e., a population bottleneck, would suggest there were multiple haplotypes carrying the resistant variant(s) in the parental population. While field populations of parasitic nematodes are expected to have an abundance of genetic diversity for selection to act upon [50,59], this finding in a small subset of individuals (200 $L_3$ per line) from an inbred laboratory isolate, emphasises the incredible capacity of these organisms to rapidly adapt to prevailing conditions.

**Table 6. Differentially expressed genes shared in all four pairwise comparisons at adjusted P<0.01.**

| | CTL vs IVM males | | CTL vs MOX males | | CTL vs IVM females | | CTL vs MOX females | |
|---|---|---|---|---|---|---|---|---|
| Gene ID | log2FC | padj | log2FC | padj | log2FC | padj | log2FC | padj |
| HCON_00158160 | 8.37 | 4.26E-10 | 9.14 | 1.86E-12 | 6.78 | 3.01E-08 | 6.10 | 5.02E-07 |
| HCON_00158150 | 4.52 | 1.08E-05 | 5.42 | 5.64E-09 | 5.56 | 0.0006752 | 4.93 | 0.0021159 |
| HCON_00152760 | 3.44 | 9.02E-06 | 3.45 | 4.31E-06 | 2.18 | 1.06E-05 | 2.17 | 2.74E-06 |
| HCON_00152750 | 2.91 | 3.37E-18 | 2.86 | 1.04E-17 | 1.93 | 0.0006180 | 1.89 | 0.0003109 |
| HCON_00155465 | 2.64 | 3.56E-20 | 2.75 | 4.93E-22 | 1.78 | 1.56E-08 | 1.75 | 8.07E-09 |
| HCON_00155380 | 1.92 | 1.92E-09 | 2.10 | 1.20E-11 | 2.48 | 4.47E-19 | 2.40 | 3.77E-18 |
| HCON_00152530 | 1.68 | 0.0006306 | 1.57 | 0.0012708 | 1.25 | 0.0056859 | 1.12 | 0.0088758 |
| HCON_00156060 | -0.81 | 4.60E-05 | -0.84 | 7.89E-06 | -0.66 | 0.0032016 | -1.02 | 2.77E-09 |
| HCON_00155770 | -0.95 | 0.0099382 | -1.03 | 0.0011996 | -0.83 | 0.0071228 | -1.14 | 1.97E-06 |
| HCON_00155640 | -1.37 | 1.08E-05 | -1.44 | 1.01E-06 | -1.29 | 0.0071870 | -1.85 | 4.53E-07 |
| HCON_00155240 | -1.84 | 5.61E-10 | -2.42 | 1.04E-17 | -1.51 | 0.0003842 | -3.50 | 3.19E-25 |

## Limitations of the study

Limitations of the study include the use of pooled sequencing data throughout, so genotypes of individual susceptible and resistant worms could not be determined. The experimental selection design is relevant for subtherapeutic treatment of adult worms (underdosing) but does not replicate field dynamics, where incoming larvae will also be exposed to subtherapeutic concentrations of long acting anthelmintics (tail selection). *Haemonchus contortus* is a tractable parasitic nematode model, and findings from these experiments have direct relevance for anthelmintic treatment of closely related helminths of live-stock, but it is currently unknown how generalisable the results are for more distant species, such as the filarial nematodes

## Conclusion

Our findings highlight the rapid selection for anthelmintic resistance with subtherapeutic treatment and implicate the pre-existence of resistance haplotypes in a drug-naïve population of *H. contortus*. We demonstrate that ivermectin selected lines show emerging moxidectin resistance, underpinned by a shared genetic locus of resistance, with the same haplotypes under selection by both anthelmintics. However, striking differences in the resistance phenotype between iver-mectin and moxidectin selected lines are apparent, which may relate to differences in the inheritance of resistance within this shared locus. Further work to determine the genetic variants underlying resistance to ivermectin and moxidectin is essential to inform sustainable use of these important drugs in parasitic helminth populations.

## Supporting information

**S1 Fig. Summary of selection experiment.**
(DOCX)

**S2 Fig. Faecal egg counts (FEC) from three generations of low dose anthelmintic selection.** Treatment was on day 28 post infection.
(DOCX)

**S3 Fig. Larval development assays for benzimidazole (thiabendazole, TBZ) and levamisole (tetramisole hydro-chloride, TET).** Figure titles refer to the drug-selected lines that were subjected to the assays.
(DOCX)

**S4 Fig. Faecal egg counts (FEC) after MOX treatment (1/100th dose) of fourth generation adults of IVM-selected or CTL lines.** Treatment was on day 28 post infection.
(DOCX)

**S5 Fig. Faecal egg counts (FEC) after treatment of fourth generation adults with a half dose of IVM or MOX.** Treatment was on day 28 post infection.
(DOCX)

**S6 Fig. Nucleotide diversity ($\theta\pi$) along the genome in all samples calculated in 100 kb windows using a subsa-mpled depth of 57X.** (A-C) Ratio of nucleotide diversity in three generations (F1 to F3) of sub-therapeutic selection in comparison to F0 sample population. Key: C = control, I = ivermectin, M = moxidectin. L1 to L3 indicate biological replicate selection lines. Pi ratio = Normalised Sample $\theta\pi$/ Normalised F0 $\theta\pi$. Nucleotide diversity for each sample window was normalised by dividing by the genome-wide median diversity. Colour indicates the rank within the treatment group, where blue indicates a higher diversity than F0, and red a lower diversity. Note that the rank indicates the genome-wide ranking. No sequencing data was available for F1_I_L1. (D) Nucleotide diversity ($\theta\pi$) of the F0 sample along all chromosomes. More negative values indicate lower diversity.
(DOCX)

**S7 Fig. Reference allele counts of F3 selected samples along Chromosome V.** Using a 57X subsampled input file, SNPs were identified which reduced in the reference allele count from the F0 to the F3 generation (F0 > F1 > F2 > F3) in selected lines, but did not do so in control lines. In addition, SNPs were identified where the reference allele was zero in any F3 selected sample, but >20 in the F0 sample and F3 control samples. (A) Ivermectin selected F3 samples, (B) Moxidectin selected F3 samples. Key to colours: Blue - reference allele reduces in all selected lines of either ivermectin (IVM1 to IVM3) OR moxidectin (MOX1 to MOX3) from F0 to F3, Orange - reference allele reduces in all ivermectin AND moxidectin lines from F0 to F3, Red - reference allele count is zero in at least one F3 sample of the ivermectin (A) OR moxidectin (B) lines but it is at least 20 in the F0 and F3 control line samples.
(DOCX)

**S8 Fig. Haplotype frequencies calculated for individual microhaplotypes for two single-copy genes.** (A) *HCON_00155390 (Hc-cky-1)*, Chr V:37,487,982–37,497,398 (B) *HCON_00158050* (a serpentine receptor, class E), Chr V:41,309,214–41,318,844. Colours denote samples. Haplotypes are spread along the x-axis, with the mid position of each microhaplotype (based on start and end SNP) provided. Only haplotype frequencies >0.01 are shown.
(DOCX)

**S9 Fig. Normalised read counts for HCON_00155390 (*cky-1*) in male and female adults.** Read counts were normalised by size factor as per the DESEQ2 model.
(DOCX)

**S1 Table. Sample details for ENA.**
(XLSX)

**S2 Table. Sample data for ENA.**
(XLSX)

**S3 Table. Larval development assays on thiabendazole and tetramisole plates.**
(XLSX)

**S4 Table. Negative and significant SNPs from allele frequency model.**
(XLSX)

**S5 Table. Differentially expressed genes in IVM and MOX selected male and female adults versus controls.** *C. elegans* homologues and Interpro domains are listed.
(XLSX)

## Author contributions

**Conceptualization:** Jennifer McIntyre, Eileen Devaney, James A. Cotton, Collette Britton, Ray M. Kaplan, Dave Bartley, Roz Laing.

**Data curation:** Jennifer McIntyre.

**Formal analysis:** Jennifer McIntyre, James A. Cotton, Roz Laing.

**Funding acquisition:** Roz Laing.

**Investigation:** Alison Morrison, Kirsty Maitland, Dave Bartley, Roz Laing.

**Methodology:** Jennifer McIntyre, Alison Morrison, Kirsty Maitland, James A. Cotton, Ray M. Kaplan, Roz Laing.

**Project administration:** Roz Laing.

**Resources:** Alison Morrison, Dave Bartley.

**Visualization:** Jennifer McIntyre.

**Writing – original draft:** Jennifer McIntyre, Roz Laing.

**Writing – review & editing:** Jennifer McIntyre, Alison Morrison, Eileen Devaney, James A. Cotton, Collette Britton, Ray M. Kaplan, Roz Laing.

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
