## [Decision Letter · Decision Letter 0]

23 Jul 2025

PPATHOGENS-D-25-01047

Analyses of emerging macrocyclic lactone resistance: speed and signature of ivermectin and moxidectin selection and evidence of a shared genetic locus

PLOS Pathogens

Dear Dr. Laing,

Thank you for submitting your manuscript to PLOS Pathogens. After careful consideration, we feel that it has merit but does not fully meet PLOS Pathogens's publication criteria as it currently stands. Therefore, we invite you to submit a revised version of the manuscript that addresses the points raised during the review process.

Please submit your revised manuscript within 30 days Sep 21 2025 11:59PM. If you will need more time than this to complete your revisions, please reply to this message or contact the journal office at plospathogens@plos.org. Please include the following items when submitting your revised manuscript:

We look forward to receiving your revised manuscript.

Kind regards,

Richard J. Martin, BVSc, PhD, DSc, DipECVPT, FRCVS

Guest Editor

PLOS Pathogens

Edward Mitre

Section Editor

PLOS Pathogens

Sumita Bhaduri-McIntosh

Editor-in-Chief

PLOS Pathogens

orcid.org/0000-0003-2946-9497

Michael Malim

Editor-in-Chief

PLOS Pathogens

orcid.org/0000-0002-7699-2064

**Additional Editor Comments :**

Your manuscript has been reviewed by 3 expert referees who find that it is written well, with the main conclusions well supported. The manuscript using ivermectin and moxidectin is seen to highlight the damaging effects of under dosing of anthelmintics. They do however, point out a number of concerns and comments that should be addressed before publication can be accepted. Accordingly you are requested to address each of their points in a response to the referees and to submit an appropriately revised manuscript.

**Journal Requirements:**

At this stage, the following Authors/Authors require contributions: Jennifer McIntyre, Alison Morrison, Kirsty Maitland, Eileen Devaney, James A. Cotton, Collette Britton, Ray M. Kaplan, Dave Bartley, and Roz Laing. Please ensure that the full contributions of each author are acknowledged in the "Add/Edit/Remove Authors" section of our submission form.

3) We have noticed that you have uploaded Supporting Information files, but you have not included a complete list of legends. Please add a full list of legends for your Supporting Information files (Supplementary Tables) after the references list.

4) Thank you for indicating that "the data for this study have been deposited in the European Nucleotide Archive (ENA) at EMBL-EBI under accession number PRJEB88928 (https://www.ebi.ac.uk/ena/browser/view/PRJEB88928). Please note that, though access restrictions are acceptable now, your entire minimal dataset will need to be made freely accessible if your manuscript is accepted for publication. This policy applies to all data except where public deposition would breach compliance with the protocol approved by your research ethics board. 

5) Thank you for stating "The funder had no involvement in the study design, collection, analysis and interpretation of data, writing of the manuscript or in the decision to submit the article for publication." Please modify this to the standard : "The funders had no role in study design, data collection and analysis, decision to publish, or preparation of the manuscript.".

**Reviewers' Comments:**

Reviewer's Responses to Questions

**Part I - Summary**

Reviewer #1: This manuscript explores the relationship between resistance to two macrocyclic lactone anthelmintics, ivermectin and moxidectin. Although moxidectin’s increased efficacy against ivermectin-resistant animals has been well documented, the current work explores how selection for resistance to each drug, caused by underdosing, affects the efficacy of both drugs against a population. Generally, I believe the work is well done by a group of authors who have continued to push our understanding of ML resistance in Haemonchus. However, I have one major comment and multiple smaller comments that should be addressed for more detail and clarity to the information being presented.

Reviewer #2: This study reports the first body of work that explores the development of anthelmintic resistance 'in real time' and combines well designed experimental infections with complex genome-wide analyses. As such it represents a significant contribution to the field. With a focus on the macrocylic lactones, ivermectin and moxidectin, its relevance extends beyond the veterinary sector to encompass human health. It is a valuable study that highlights the detrimental impact of sub-therapeutic doses of anthelmintics.

Reviewer #3: In McIntyre et al, the authors demonstrate that even single, subtherapeutic doses of ivermectin or moxidectin rapidly select for preexisting resistance alleles in Haemonchus contortus, pinpointing a shared hotspot on chromosome V. By subjecting a drug susceptible isolate to successive low dose treatments, they show that three rounds of underdosing with ivermectin are sufficient to confer full clinical resistance, and cross resistance to moxidectin, whereas moxidectin underdosing produces a strong genomic signature without equivalent phenotypic protection. These findings provide the first direct evidence that underdosing can swiftly drive anthelmintic resistance and identify a genetic marker for early surveillance and intervention. The manuscript is well written and appears sound, and the main conclusions are well supported. Before acceptance the following comments/concerns need to be addressed.

1. RNA-seq analysis. Post-treatment adult samples were used in the RNA-seq analysis. It is unclear when these adult worms were collected. Were they collected 14 days after treatment? Several genes were found to be downregulated in the selected lines (Fig. 7 and Table 6). Given the high SNP density, which is likely to be even higher in the selected lines at the selected locus compared to controls, RNA-seq reads may have been too divergent from the reference genome for the aligner to map them accurately. This could result in lower read counts. HISAT2, when used with default settings, cannot accommodate a large number of mismatches per read. Based on our experience, STAR performs better than HISAT2 for aligning divergent reads. STAR also allows adjustment of mismatch tolerance (--outFilterMismatchNmax). Only a small number of genes were differentially expressed across all four pairwise comparisons (Table 6). However, no functional descriptions are provided for these genes. Do their predicted functions offer any insight into possible mechanisms of resistance?

2. Analysis of longitudinal allele frequency change (Fig. 6 and Supplementary Fig. 6). This analysis implicitly assumes that the reference allele represents the susceptible allele. While this may be true for most SNPs, it might not hold for all cases. It would be useful to explicitly confirm whether the major allele in the F0 population corresponds to the reference allele. If this assumption does not hold for a substantial number of SNPs, the analysis could be revised using the major allele in F0 rather than using the reference allele.

3. Have you attempted fine mapping using individual SNP CMH p-values or FST values? In the moxidectin analysis, did the authors observe a large number of individual SNPs with FST equal to 1? If so, this could make it difficult to further narrow down the selected locus using pool-seq data alone. In Fig. 6, a small number of individual SNPs show highly negative slope coefficients. Can the authors clarify whether these signals are biologically meaningful or likely to result from technical noise?

4. The conclusion states that the findings of this study implicate pre-existence of “multiple” ivermectin and moxidectin resistance haplotypes in a drug-naïve population of H. contortus (line 704), but this claim lacks direct supporting evidence. With the available pool-seq data, it may be possible to investigate this by analyzing microhaplotypes (short stretches of DNA containing multiple linked SNPs). Several methods allow microhaplotype inference from Pool-seq data (e.g., PMID: 37923981). Analyzing the selected locus in this way could reveal whether multiple non-reference haplotypes were selected and whether the same haplotypes appear under both ivermectin and moxidectin selection.

5. The manuscript suggests that moxidectin resistance may be “more multigenic” than ivermectin resistance. However, the current findings are also fully consistent with a single gene model involving a recessive trait. The data do not directly support one interpretation over another. Please clarify that the proposed multigenic architecture is speculative and not clearly supported by the present data.

6. The authors could expand the discussion of how these findings apply to deworming practices in endemic areas. Additionally, limitations of the study should be stated more clearly. For example, it would be useful to comment on how well the experimental selection reflects real-world field dynamics, and whether these results are generalizable to O. volvulus.

7. Please review the “Treatment Efficacy” column in Tables 3 and 4 to ensure correct number formatting.

8. Given the broad readership of the journal, it would be helpful to briefly define or explain the following terms or acronyms at their first appearance, or where appropriate:

- Genetic basis

- Genetic architecture

- QTL

- FECR

- ED50

9. Currently, only the BioProject ID is provided, individual sample accession numbers should be included in the supplementary information.

10. Supplementary Fig. 2. There is a mismatch between the legend and the figure content. The legend refers to TBZ and TET, while the figure labels IVM and MOX.

11. For Supplementary Fig. 7, the authors should clarify how read counts were normalized. Was normalization based on library size, and can the units be more descriptive to reflect the normalization used?

**Part II – Major Issues: Key Experiments Required for Acceptance**

Reviewer #1: Major comments

The idea of multi-drug resistance genes is repeated throughout the manuscript, and noted that none were observed in the current work. The authors mention detox pathways and P-glycoproteins as potential MDR mechanisms. On the first mention, the authors make the distinction that upregulation of these proteins “might” be expected. However, they seem to double down on the concept further in the manuscript. Although such pathways “could” be involved in MDR, no involvement has been validated at the molecular level. On the contrary, for the pathways that have been somewhat explored, such proteins are found to interact with specific classes, or in at least one case, of a single drug within the class. I suggest that the authors be more forthcoming that current data could argue against MDR-associated pathways and at a minimum acknowledge that either direction, broad or specific mechanisms, could be equally possible.

Minor comments

I am generally aware of the history of the MHco3(ISE) isolate, but I think at a minimum, the information stating that it is fully susceptible should have relevant citations. Additionally, has the isolate been periodically phenotyped to confirm 100% susceptibility to common anthelmintics? Lastly, has the current stock of the isolate been confirmed to be molecularly pure H. contortus?

Line 95-99: There has not been confirmed validation of a gene associated with IVM/MOX resistance, and cky-1 (found through BSA) has also not been validated as the causal gene. My question is what evidence supports that the animals in the cited work were truly heterozygous vs homozygotes at relevant loci and how does this change the interpretation of the concept being presented?

Line 180: Were the bags simply sealed in ziplock bags as opposed to vacuum sealing? How long were the plates stored? Both could impact the activity of the drug if the plates were stored for extended lengths of time or in a way that could allow some desiccation.

Line 187-191: Were the plate concentrations of drugs made with knowledge that 40 µL of additional volume would be added? With the addition of the embryos and nutritive media, the concentration of drugs in the well would have significantly decreased after absorption into the agar, impacting the results. If concentrations were not adjusted for this dilution, actual exposure concentrations should be calculated and the data should be interpreted given the actual concentrations.

Line 233: It appears that you discuss how indels were called in the sequences, but I do not see how SNVs were called in the methods, although below they are discussed as part of the analysis for variants. Please add these details.

Line 245: To only cover 31.7% of the total bases in the assembly leads me to believe that there are elements of diversity that would be missed in most of the genome and not factored into the current analysis. Was there a reason that only one third of the genome was considered instead of doing further sequencing?

Line 294: I assume the window on chromosome V was chosen due to the previous data indicating an interval containing cky-1 as a candidate gene for ML resistance, but the previous work is not mentioned to justify why this window was chosen. I see that the first mention of this work is at line 465, but please discuss prior to this point the previous work so that this window is justified in context.

Line 422-425: As a Discussion point, consider briefly presenting the implications of how quickly ivermectin resistance would become highly moxidectin-resistant in a field population.

Figure 2: If axis labels are going to be used for all the plots in the figure, please ensure that the labels are cleanly shown for all plots, as currently the genomic position seems to be specific for plot D, but applies to all plots. I also suggest adding more space between your lettering for each plot and the y-axis labels.

Figure 3 and other figures: This may be an effect of the upload for the draft, but please ensure all figures are of publication quality. Currently, some figures are grainy/pixelated in the draft. I also suggest adding more space between your lettering for each plot and the y-axis labels, and this issue is present in multiple figures.

Line 644-645: Are any of the DEGs known to be involved in drug response or could make sense in the context of ML resistance?

Table 6. I suggest that, where possible, specific gene names (or C. elegans orthologs) be listed. The HCON_**** nomenclature is not informative without actually having to search for each gene. Additionally, the HCON_00158160 gene ID does not appear in a search of the genome. Please confirm that the listed IDs are accurate.

Specific comments

Line 145-149: I suggest separating the very long sentence about dosing into two separate doses. Currently, it comes off somewhat confusing with the “then ivermectin… and .008 mg/kg…”

Line 186: Please cite a reference for “standard procedures”.

Line 405: The use of recapitulated for something that has not previously been validated seems like awkward language. I suggest simply saying that MDR mechanisms were not observed.

Reviewer #2: None required

Reviewer #3: (No Response)

**Part III – Minor Issues: Editorial and Data Presentation Modifications**

Reviewer #1: (No Response)

Reviewer #2: The manuscript was generally well written but at times the use of 'three third generation' and similar phrases a bit clumsy. As the narrative continues there is use of F0, F1, F2, F3 which was much easier to digest - perhaps these could be introduced earlier e. g 3 x F3 rather than 'three third generation'.

The experimental work in animals is well explained but is complex, I think an inclusion of an workflow (as supplementary?) or infographic showing these experiments would help with understanding. Given the extensive and well conducted molecular and genomic work is carried out on these F0-F4 populations it is crucial that this is well explained and I think a workflow/infographic of the sheep experiments would really help.

The larval assays indicated that moxidectin resistance demonstrate side resistance to ivermectin, it is a shame that there is not in vivo data also showing this, for example treatment of IVM-selected F3 with MOX in vivo, and vice versa. These experiments were not done, but given the work involved I am not proposing they must be, rather just a mention of the that this might have been an expected outcome had they been done.

There were a couple of results that were intriguing that could perhaps be discussed a little further:

1. The apparent rapid selection of resistance seen by the FECRT data by both MOX and IVM following just one treatment with a subtherapeutic dose is somewhat surprising. How does this observation align with the development of resistance in naturally infected sheep and cattle? For example over what time frame has resistance developed in Haemonchus populations. I am just wondering, despite the ISE isolate still displaying substantial genetic diversity if these pre-existing alleles are at higher frequency in the F0 population than in WT populations of Haemonchus. Perhaps a line to discus this?

2. Given there are the two sexes and known differences in expression patterns between males and females, what attempts were made with the 5000 L3 and the batches of 200L3 to ensure they represented a mix/balance of the two sexes. Is there any data, or easily accessible method, of determining the species mix of the larvae?

Reviewer #3: (No Response)

PLOS authors have the option to publish the peer review history of their article (what does this mean?). If published, this will include your full peer review and any attached files.

Reviewer #1: No

Reviewer #2: No

Reviewer #3: No

**Figure resubmission:**
---

## [Editor Report · Decision Letter 1]

29 Sep 2025

Dear Dr Laing,

We are pleased to inform you that your manuscript 'Analyses of emerging macrocyclic lactone resistance: speed and signature of ivermectin and moxidectin selection and evidence of a shared genetic locus' has been provisionally accepted for publication in PLOS Pathogens.

Best regards,

Richard J. Martin, BVSc, PhD, DSc, DipECVPT, FRCVS

Guest Editor

PLOS Pathogens

Edward Mitre

Section Editor

PLOS Pathogens

Sumita Bhaduri-McIntosh

Editor-in-Chief

PLOS Pathogens

orcid.org/0000-0003-2946-9497

Michael Malim

Editor-in-Chief

PLOS Pathogens

orcid.org/0000-0002-7699-2064

The manuscript contains new and impactful studies on the development of resistance to ivermectin and moxidection associated with under dosing.The authors have responded positively to the comments of 3 expert reviewers.
---

## [Editor Report · Acceptance letter]

Dear Dr Laing,

We are delighted to inform you that your manuscript, "Analyses of emerging macrocyclic lactone resistance: speed and signature of ivermectin and moxidectin selection and evidence of a shared genetic locus," has been formally accepted for publication in PLOS Pathogens.

Best regards,

Sumita Bhaduri-McIntosh

Editor-in-Chief

PLOS Pathogens

orcid.org/0000-0003-2946-9497

Michael Malim

Editor-in-Chief

PLOS Pathogens

orcid.org/0000-0002-7699-2064